# Non-compliance with clinical guidelines increases the risk of complications after primary total hip and knee joint replacement surgery

Helen Mary Badge[1,2,3,4☯]*, Tim Churches[2,3☯], Justine M. Naylor[1,2,3☯], Wei Xuan[2,3☯], Elizabeth Armstrong[5‡], Leeanne Gray[6‡], John Fletcher[7,8‡], Iain Gosbell[3,9‡], Christine Lin[10‡], Ian A. Harris[1,2,3,6☯]

1 Whitlam Orthopaedic Research Centre, Liverpool, Australia, 2 South Western Sydney Clinical School, UNSW, Liverpool, Australia, 3 Ingham Institute for Applied Medical Research, Liverpool, Australia, 4 Australian Catholic University, North Sydney, Australia, 5 Falls Balance and Injury Research Centre, Neuroscience Research Australia, Randwick, Australia, 6 South Western Sydney Local Health District, Liverpool, Australia, 7 University of Sydney, Camperdown, NSW, Australia, 8 Westmead Hospital, Westmead, NSW, Australia, 9 Western Sydney University, Campbelltown, NSW, Australia, 10 Sydney School of Public Health, The University of Sydney, Camperdown, NSW, Australia

☯ These authors contributed equally to this work.
‡ These authors also contributed equally to this work.
* helen.badgehawke@gmail.com

**Data Availability Statement:** A deidentified version of the data set and the full R code for all analyses

## Abstract

### Background

Total hip and total knee replacement (THR/TKR) are common and effective surgeries to reduce the pain and disability associated with arthritis but are associated with small but significant risks of preventable complications such as surgical site infection (SSI) and venous-thrombo-embolism (VTE). This study aims to determine the degree to which hospital care was compliant with clinical guidelines for the prevention of SSI and VTE after THR/TKR; and whether non-compliant prophylaxis is associated with increased risk of complications.

### Methods and findings

A prospective multi-centre cohort study was undertaken in consenting adults with osteoarthritis undergoing elective primary TKR/THR at one of 19 high-volume Australian public or private hospitals. Data were collected prior to surgery and for one-year post-surgery. Four adjusted logistic regression analyses were undertaken to explore associations between binary non-compliance and the risk of surgical complications: (1) composite (simultaneous) non-compliance with both (VTE and antibiotic) guidelines and composite complications [all-cause mortality, VTE, readmission/reoperation for joint-related reasons (one-year) and non-joint-related reasons (35-days)], (2) VTE non-compliance and VTE outcomes, (3) antibiotic non-compliance and any SSI, and (4) antibiotic non-compliance and deep SSI. Data were analysed for 1875 participants. Guideline non-compliance rates were high: 65% (VTE), 87% (antibiotics) and 95% (composite guideline). Composite non-compliance was not associated

are available (https://doi.org/10.26190/mddw-by48).

**Funding:** The study was funded via a grant through the HCF Health and Medical Research Foundation. HB is supported by an Australian Government Research Training Program Scholarship. CL is supported by a fellowship from the National Health and Medical Research Council, Australia. Grant Number: IHIIAMR2012073043 The funders had no role in study design, data collection and analysis, decision to publish, or preparation of the manuscript.

**Competing interests:** JF is an unpaid member of the group that developed the NHMRC Guidelines for the prevention of VTE. This does not alter our adherence to PLOS ONE policies on sharing data and materials.

**Abbreviations:** AOR, Adjusted odds ratio; DVT, deep vein thrombosis; NHMRC, National Health and Medical Research Council; PE, pulmonary embolism; SSI, surgical site infection; TGAb, Therapeutic Guidelines Antibiotic; THR, Total hip replacement; TKR, total knee replacement; VTE, venous-thrombo-embolism.

with composite complication (12.8% vs 8.3%, adjusted odds ratio [AOR] = 1.41, 95%CI 0.68–3.45, p = 0.40). Non-compliance with VTE guidelines was associated with VTE outcomes (5% vs 2.4%, AOR = 2.83, 95%CI 1.59–5.28, p < 0.001). Non-compliance with antibiotic guidelines was associated with any SSI (14.8% vs 6.1%, AOR = 1.98, 95%CI 1.17–3.62, p = 0.02) but not deep infection (3.7% vs 1.2%, AOR = 2.39, 95%CI 0.85–10.00, p = 0.15).

## Conclusions

We found high rates of clinical variation and statistically significant associations between non-compliance with VTE and antibiotic guidelines and increased risk of VTE and SSI, respectively. Complications after THR/TKR surgery may be decreased by improving compliance with clinical guidelines.

## Background

Primary elective total hip replacement (THR) and total knee replacement (TKR) are effective surgical procedures that reduce pain and disability associated with severe arthritis [1, 2]. In 2019, over 97,000 THR and TKR were performed in Australia [3] and nearly 232,000 in the United [4]. Demand for these procedures continues to grow [5, 6]. Although these procedures are cost-effective, they are associated with a small but important risk of complications that increase mortality, morbidity and cost [1, 7].

THR and TKR patients are considered at risk of venous thromboembolism (VTE) and surgical site infection (SSI), both of which are associated with poorer patient experience, high disease burden and increased costs for patients and the health system [8–10]. There is strong evidence that prophylaxis for VTE and SSI after THR and TKR is effective [11, 12]. Compliance with the available evidence-based clinical guidelines [13–15] is thought to be low and contributes to unwarranted variation in current VTE and antibiotic prophylaxis between surgeons and hospitals [16–18]. Although patient care may need to be varied from recommended care to address patient-specific issues, unwarranted variation may increase costs and negatively impact patient outcomes and service capacity [19, 20]. Consequently, programs to improve compliance with clinical guidelines are being implemented internationally due to the potential to improve the value of THR and TKR [21–23].

Despite clinical guidelines being evidence-based, prospective studies have not explored the association between compliance with recommended care and complications. Using a prospective cohort of participants who underwent elective TKR or THR in Australian hospitals, this study aims to determine the magnitude of guideline non-compliance and answer the following questions:

1. Is there an association between simultaneous non-compliance with both (antibiotic and VTE) guidelines and the rate of surgical complications after elective primary total joint replacement surgery?

2. Is there an association between non-compliance with VTE prophylaxis guidelines and VTE after elective primary total joint replacement surgery?

3. Is there an association between non-compliance with antibiotic guidelines and postoperative infection rate (considering all SSI and deep SSI requiring readmission/reoperation) after elective primary total joint replacement surgery?

## Methods

### Registration and data collection

A prospective observational cohort study of people undergoing elective primary total hip or knee replacement for osteoarthritis in one of 19 high-volume institutions in Australia was performed to examine the relationship between non-compliance with nationally recommended contemporaneous VTE prophylaxis guidelines and antibiotic guidelines and patient outcomes. Eligible sites included private and public Australian hospitals with high annual surgical volume (over 275 cases per year) of THR and TKR surgery. Inclusion criteria for participants in the study were: consenting adults (over 18 years) with a primary diagnosis of osteoarthritis undergoing primary TKR or THR; sufficient English to comprehend the protocol; and available to participate in follow-up for 12 months.

Investigators identified 36 eligible sites, including 27 identified through random selection from eligible sites listed on the Australian 'My Hospitals' website and an additional nine sites identified by convenience sampling [24]. Twenty sites elected to participate, but one was denied governance approval leaving 19 participating sites. Sites were provided with face-to-face and remote training, study resources, ongoing support, and reimbursement for each eligible participant with complete acute data. Site coordinators screened and recruited participants during routine pre-admission assessment. Consecutive screening of all potential participants was planned although staff absences interrupted screening at some sites. Prior to commencement, the study protocol was registered (NCT01899443) [25] and ethical approval was obtained from nine human research ethics committees.

Prior to data collection informed written consent was obtained from eligible participants and the signed consent form was witnessed by the site coordinator. The site coordinators collected prospective pre-operative data from participants via interview, including socio-demographic information, past medical history, indications, and contraindications for VTE and antimicrobial prophylaxis, and acute care data from the medical records. Participants provided post-acute data via telephone follow-up at approximately 35, 90, and 365 days post-surgery. Participants and sites provided details regarding prophylaxis and surgical complications.

The researchers completed an audit of all medical records and by contacting surgeons, primary care physicians and other hospitals to verify the accuracy of patient-reported and acute complications. Any reported complication was coded as a dichotomous variable to indicate whether the participant did or did not experience the complication. The primary outcome was a composite outcome comprising all-cause mortality, any VTE and any reoperation or readmissions within 35 days for medical issues or within 365 days for joint-related complications. Secondary outcomes included any VTE event [pulmonary embolism (PE) or deep vein thrombosis (DVT)], any SSI (requiring oral or IV antibiotics, readmission, or reoperation) and only *deep* SSI (requiring readmission or reoperation) up to 365 days post-surgery.

### Criteria for compliance with clinical guidelines

Compliance was calculated with the recommendations of two nationally produced guidelines for Australian health services:

i.  National Health and Medical Research Council (NHMRC) Clinical Practice Guideline for the Prevention of Venous Thromboembolism (Deep Vein Thrombosis and Pulmonary Embolism) in Patients admitted to Australian hospitals (2009) [13]; and

ii.  Therapeutic Guidelines: Antibiotic Version 14 (2010) [14]

These guidelines were current during the study. Discrete elements of compliance were identified from the recommendations in each guideline. Investigators engaged in an *a priori* iterative consensus process to determine clear criteria for compliance versus non-compliance with each element of care compliance that could be consistently applied to assess the variety of prophylactic regimens (See Table 1). This process was vital where recommendations were ambiguous or hard to define and allowed patient appropriate deviation to be considered compliant.

The study criteria for VTE compliance for the appropriate prophylactic agent and dose were more lenient than the guideline recommendations. The NHMRC recommendations were ambiguous about the use of warfarin in relation to the concurrent management of cardio-vascular disease [13]. We allowed (as 'compliant') the use of warfarin or UFH and higher than

**Table 1. Criteria for compliance with NHMRC guidelines for prevention VTE (2009) [13] and therapeutic guidelines antibiotics (2010) [14].**

| Criteria for compliance: | VTE prophylaxis |
|---|---|
| 1. Right drug | *If no contraindications o*ne or more recommended drugs: Low molecular weight heparins (*LMWH*—enoxaparin or dalteparin*), fondaparinux, rivaroxaban, dabigatran etexilate; warfarin / unfractionated heparin (UFH).* Use of other drugs, for example aspirin was ignored in determining compliance. |
| 2. Right dose | ≥40mg enoxaparin sodium (or ≥20mg if renal impairment); ≥5000u dalteparin (or ≥2500u if renal impairment), ≥2.5mg fondaparinux, ≥10mg rivaroxaban, ≥150mg dabigatran etexilate, any dose warfarin or UFH. Every dose of all recommended drugs used must be at the dose recommended in the guidelines or higher |
| 3. Right duration of chemical prophylaxis | At least one recommended drug commences on Day 0 (day of surgery) or Day 1 AND continues for at least the minimum recommended duration for that drug, with no more than 2 missed days for any reason. The recommended duration is ≥ 27 days for THR and ≥ 9 days for TKR. Any duration was considered compliant if VTE was diagnosed within this period. |
| 4. Right mechanical device/s used | If eligible for chemical and mechanical prophylaxis, THR to use at least one of: foot pumps, calf compressors, or graduated compression stockings (GCS). TKR to use at least one of either foot pumps or calf compressors. |
|  | If not eligible for chemical prophylaxis, both THR and TKR to use two mechanical devices, unless contraindicated. |
|  | If not eligible for either chemical or mechanical prophylaxis can use or not use any device without penalty. |
| VTE compliance | Compliant with 1 AND 2 AND 3 AND 4 |
| Criteria for compliance: | Antibiotic Prophylaxis |
| 5. Right drug | Only cefazolin OR flucloxacillin OR vancomycin (only if indicated*) |
| 6. Right dose | 1g (or 2g if ≥ 80kg) cefazolin; 2g flucloxacillin; 25mg/kg up to 1.5g (≥ 60kg = 1.5g) vancomycin. If surgery time exceeds 3 hours or is continuing 4 hours after first dose, patients should receive a second dose of cefazolin or flucloxacillin intra-operatively. |
| 7. Right pre-op timing | The first dose of any antibiotic is given at the time of induction (up to any time prior to skin incision). We ignored the recommendation re commencing antibiotics when tourniquet used in TKR at if given within 5 mins prior to tourniquet inflation, or just before release). |
| 8. Right duration | Prophylactic antibiotics were ceased within 27 hours and if vancomycin was used only a single dose given. |
| Antibiotic compliance | Compliant with 1 AND 2 AND 3 AND 4 |
| Composite compliance | Compliant with all VTE guideline elements 1 AND 2 AND 3 AND 4, AND all Antibiotic guideline elements 1 AND 2 AND 3 AND 4 |

*Indications: allergy to penicillin, cephalosporins or all beta-lactam antibiotics, history of multi-resistant organisms, hospital admission longer than 5 days within 3 months preoperatively.

recommended doses for any recommended drug. While neither warfarin nor heparin are commonly used for primary VTE prophylaxis in Australia, this enabled clinically appropriate decisions regarding therapy for comorbid conditions to take priority over preventing a potential VTE. We could not reach agreement regarding the duration of mechanical prophylaxis based on the recommendation to use 'until regained full mobility', so compliance with mechanical prophylaxis was not included [13].

The guideline recommendations regarding antibiotic prophylaxis were less ambiguous [14]. We used patient-reported indications and contraindications to indicate when vancomycin was used appropriately but did not impose penalty for non-use of vancomycin given participants may not be accurate in reporting allergies. Regarding the criteria for compliant duration of antibiotics, we allowed 3 hours longer than the 24 hours recommended to accommodate minor variations in scheduled drug administration.

Compliance was assessed as a series of dichotomous variables for each element of the guideline, for patients who completed at least one follow-up. Prophylaxis was considered non-compliant with the overall guideline if one or more elements were considered non-compliant. Composite (overall) compliance required prophylaxis that was compliant with all elements of both the VTE and the antibiotic guidelines.

Determining compliance required assessment of complex data describing the prophylaxis received as well as patient specific indications and contra-indications. Computer-based algorithms were developed in R to automatically generate compliance results. This ensured the consistent application of the criteria for compliance and the accuracy of these data was confirmed by comparing these results with manually calculated compliance results.

## Sample size calculation

Based on a previous study [26], the *a priori* compliance to non-compliance ratio in these participants was taken to be 2:1 (i.e. 67% versus 33%), and the prevalence of the composite outcome to be 7% in the compliance subgroup (reference group). Calculation determined that 1102 participants in the compliant group and 551 participants in the non-compliant group (1653 in total) would provide 80% power at a 5% significance level to detect a significant Relative Risk (RR) of 1.6 for non-compliance. The prevalence of the composite outcome was estimated to be 11% in the non-compliant group compared to 7% in the compliant group. The planned sample size was increased to 2200 to consider a multiple correlation coefficient of 0.1 among known confounders and allow for 7% loss to follow-up by 35 days.

## Data analyses

All data were entered into a designated REDCap database hosted by the University of New South Wales [27]. Initial analysis was performed (by WX) using SAS [28]. All final analyses were conducted Independently (by TC, HB and IAH) using Foundation for Statistical Computing Platform (version 3.6.1) [29]. Descriptive statistics were calculated to profile site-level and participant-level characteristics. Results were presented as the median and inter-quartile range (IQR) or mean and standard deviation (SD). Some variables [bilateral joint, smoking status, American Society of Anesthesiology score (ASA), education, neuraxial anaesthesia] were collapsed to allow for adequate sample size or clinically meaningful groups to be included in analyses. Bivariable analyses were undertaken for each outcome.

We conducted four conducted multiple logistic regression analyses to explore associations between non-compliance and risk of outcomes as follows: (1) non-compliance with both VTE and antibiotic guidelines and composite outcomes (2) VTE non-compliance and VTE outcomes, (3) antibiotic non-compliance with any SSI and (4) antibiotic non-compliance and

deep SSI. Patient and care factors known to increase the risk of surgical complications (including VTE and SSI) were considered as potential confounders [30–32]. Factors identified on univariate analysis with a p-value < 0.25 were entered into a backwards, stepwise multivariable logistic regression model (using the Akaike information criterion–AIC) to identify the association between guidelines compliance and complication outcomes for each analysis. We reported the final model after backwards stepwise regression using AIC and forcing only the main predictor (non-compliance) into each model.

Missing data were imputed using multivariate imputation by chained equations (MICE). Model selection was performed using one of the imputed datasets, and effect estimates were taken from the pooled estimates using the five imputed datasets. We tested the models with all two-way interaction terms entered, and none of these was significant. Sensitivity analyses were performed using complete case analysis and Bayesian information criterion (BIC). Further sensitivity analyses were completed without including routine doppler ultrasound (DUS), given this may mediate VTE complication outcomes. Interaction terms for the main predictor (non-compliance) against each other variable were tested in the final model for each analysis. A de-identified version of the data set and the full R code for all analyses are available (https://doi.org/10.26190/c46r-ne05).

## Results

### Sample ascertainment

Seventy-seven percent (2529/3285) of all patients screened were eligible for participation (See Fig 1). Of these, 2143 people provided consent preoperatively, and data were received for 1905 (88.9%) consenting participants as some did not proceed to surgery or no acute data were received by investigators. Thirty (1.6%) people were excluded from analyses as they did not have any post-acute follow up. Missing data for each variable was less than 2% for all variables except ASA class (2.2% missing).

### Sites, surgeon and participant characteristics

Site, surgeon, and participant characteristics are provided in Table 2. There were 19 sites from five Australian states. Sites included 10 public hospitals that completed 45.9% of surgical procedures, the other sites being private hospitals. The number of participants ranged from 12 to 294 from each site, and 1–125 per surgeon. Routine Doppler ultrasound was performed by one site and two surgeons at two other sites.

Indications for appropriate use of vancomycin were reported for 17.2% (N = 322) of the sample, although vancomycin was used appropriately in 31.6% of the 136 of people who received it and not used in 14.1% of participants with relevant indications (See Table 3 and S1 and S2 Tables in S1 File). Almost 30% of all participants were taking pre-operative medications commonly used for postoperative VTE prophylaxis, and this included 20.4% of people who reported taking medication for comorbid heart disease (See S3 Table in S1 File).

### Participant outcomes: Surgical complications up to one year

There were 355 surgical complications that met the criteria for the composite complication that were experienced by 234 (12.5%) participants (See Table 3). Five participants died from surgical complications, and seven died from medical causes unrelated to the surgery. The incidence of VTE was 4.1% (N = 76), with nearly three-quarters of the cases being DVT alone (See Table 3 & S4 Table in S1 File). Joint-related complications in the first year after surgery accounted for 85.9% (N = 159/185) of all readmissions and 88.8% (119/134) of all reoperations

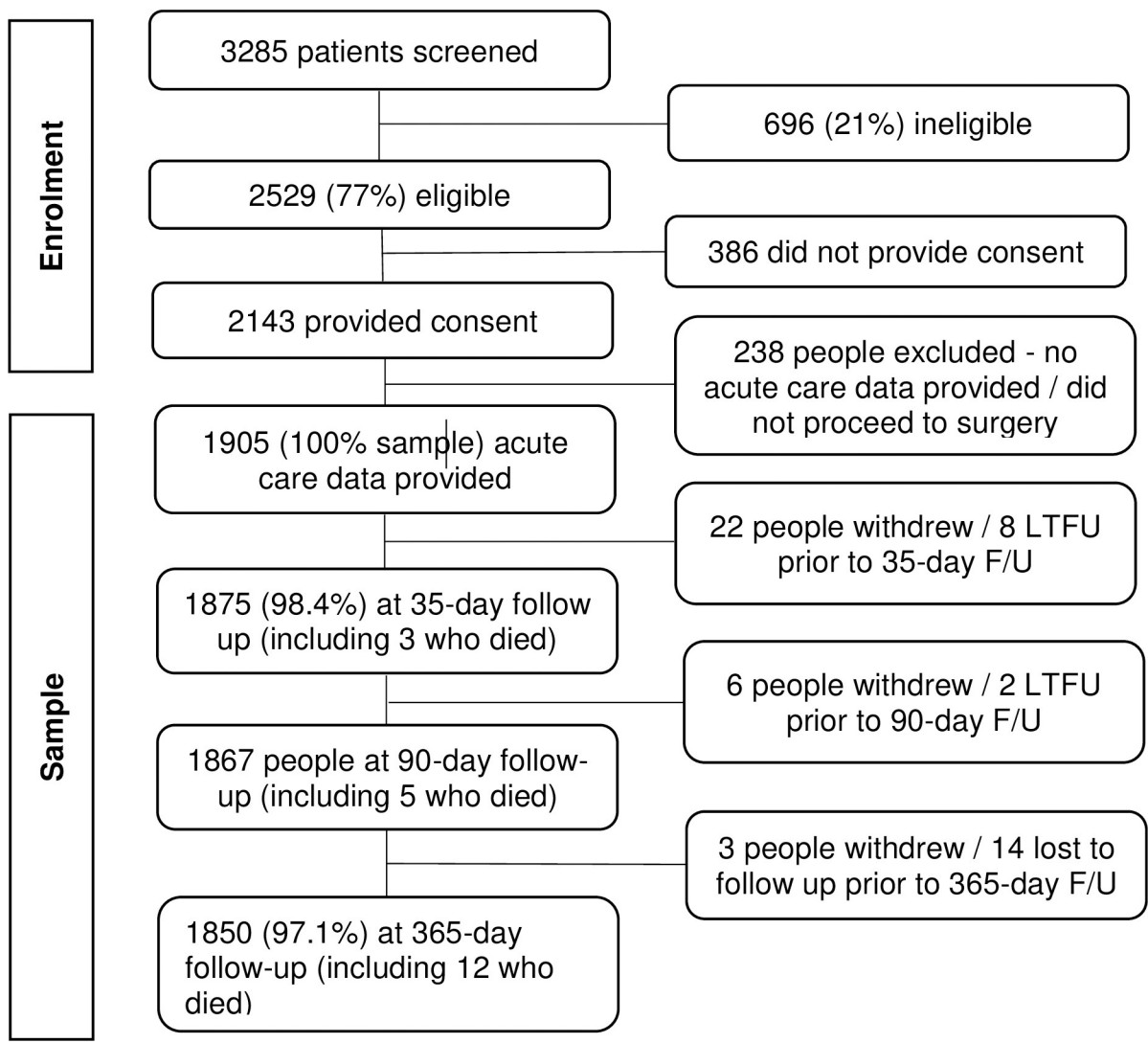

**Fig 1. Participant recruitment, eligibility, and participation results.**

(See Table 3 & S5-S7 Tables in S1 File). Deep infection included joint or wound infections requiring intravenous (IV) antibiotics or readmission (N = 46) or reoperation (N = 17) and were experienced by 3.4% (N = 63) of the participants. Bleeding complications requiring readmission/reoperation were experienced by 7 (0.37%) people for joint-related bleeding and 10 (0.53%) people for on-joint related bleeding (See S8 Table in S1 File) [33].

## VTE, antibiotic and combined (composite) compliance

Table 4 reports all compliance results. The level of compliance with both VTE and antibiotic guidelines was 4.5%. Overall compliance with all VTE recommendations was 35.3% [13]. Contrary to expectation, the use of multiple medications for postoperative VTE prophylaxis was common, with 45.5% (N = 854) of people receiving 2–4 different medications (See S2 Table in S1 File). Nearly a third of participants took a drug at higher or lower than recommended dose, and the duration of prophylaxis was longer than recommended for 33% THR and 58% TKR participants [13]. Overall compliance with antibiotic guidelines was 13.2% [14]. Fewer than

**Table 2. Site, surgeon and participant characteristics.**

| Site & surgeon Characteristics | Description, N (%), median (IRQ) | Results |
|---|---|---|
| Sites | Public | 10 (54%) |
| | Private | 9 (46%) |
| Number surgeons | | 121 |
| Number participants | Per surgeon [Median (IQR)] | 8 (16) |
| | Per site [Median (IQR)] | 70 (44) |
| Length of stay (days) | Median (IQR) | 5 (3) |
| **Participant characteristics** | **Description, N (%), median (IRQ)** | **N = 1875** |
| Joint (all surgeries) | Hip | 815 (43.5%) |
| | Knee | 1060 (56.5%) |
| Bilateral joint replacement | Hip | 10 (0.5%) |
| | Knee | 81 (4.3%) |
| Public hospital | Yes | 861 (45.9%) |
| Duration of surgery (mins) | Median (IQR) | 70 (44) |
| Age (years) | Median (IQR) | 67.6 (12.9) |
| Sex | Female | 1017 (54.2%) |
| Insurance status | Public | 821 (43.8%) |
| | Private health insurance | 980 (52.3%) |
| | Self-funded (private) | 29 (1.5%) |
| | Other insurance / compensation | 16 (0.9%) |
| | Department of Veterans Affairs | 29 (1.5%) |
| Post-school education | Up to school completion | 895 (48%) |
| status (N = 1866) | Post school qualification | 971 (52%) |
| BMI | Median (IQR) | 29.7 (7.9) |
| Current smoker | No | 1710 (91.7%) |
| (N = 1865) | Yes | 155 (8.3%) |
| Comorbid conditions | Heart disease | 474 (25.3%) |
| | History stroke | 113 (6%) |
| | Bleeding disorder | 19 (1%) |
| | Previous VTE (N = 1873) | 149 (8%) |
| | Diabetes | 306 (16.3%) |
| | Hypertension | 1142 (60.9%) |
| | High cholesterol | 702 (37.4%) |
| | Kidney disease | 63 (3.4%) |
| | Liver disease | 49 (2.6%) |
| | Current cancer (any type) | 41 (2.2%) |
| | History of any type cancer (N = 1873) | 220 (11.7%) |
| | Lung disease | 341 (18.2%) |
| | Anxiety / depression | 350 (18.7%) |
| | Mental health disorder | 22 (1.2%) |
| | Gastro-intestinal Reflux Disorder (GORD) | 486 (25.9%) |
| | Sleep apnoea | 133 (7.1%) |
| | Neurological conditions | 53 (2.8%) |
| | Musculoskeletal conditions (N = 1873) | 907 (48.4%) |
| | Any other comorbid conditions | 729 (38.9%) |

(*Continued*)

**Table 2.** (Continued)

| Site & surgeon Characteristics | Description, N (%), median (IRQ) | Results |
|---|---|---|
| Previous total joint replacement | Hip | 244 (13%) |
| | Knee | 308 (16.4%) |
| Medications taken for | Paracetamol | 1086 (57.9%) |
| osteoarthritis | Non-steroidal anti-inflammatories (NSAIDS) | 523 (27.9%) |
| | Opioids | 385 (20.5%) |
| | Antidepressant / antiepileptics | 36 (1.9%) |
| | Steroids | 6 (.3%) |
| Recommended indications for vancomycin | Self-reported allergy to penicillin, cephalosporin, or all beta-lactam ABs | 222 (11.8%) |
| | Patient history of MRSA infection / swab | 85 (4.5%) |
| | Patient history of gram-negative infection(s) | 1 (0.1%) |
| | Hospital admission with LOS > 5 days within 3 months of THR or TKR | 14 (0.7%) |
| American Association Anaesthetists (ASA) score(N = 1833) | 1 or 2 | 1246 (68%) |
| | 3 or 4 | 587 (32%) |
| Acute processes of care | Routine doppler performed (N = 1847) | 347 (18.8%) |
| | Cement fixation used (N = 1874) | 1204 (64.2%) |
| | Tranexamic acid used (N = 1868) | 1127 (60.3%) |
| | Neuraxial anaesthesia (N = 1874) | 1182 (63.0%) |
| | Intra-articular Drain (N = 1869) | 825 (44.1%) |
| | Tourniquet (only used for TKA) | 909 (48.5%) |
| | Blood transfusion (N = 1868) | 332 (17.8%) |
| Mobilisation post-surgery | First mobilised day 0 or 1 | 1395 (74.7%) |

half the sample received a single prophylactic antibiotic (46.4%), while 8.1% received 3 or 4 different antibiotics (See S1 & S2 Tables in S1 File). The most used antibiotic was cephazolin (90.7%).

**Association between composite non-compliance and composite surgical complications.** The higher incidence of composite outcome in the (overall) non-compliant group was

**Table 3. Prevalence of complications at one year included in the composite outcome.**

| Outcome | Yes |
|---|---|
| Mortality | 12 (0.6%) |
| All VTE events | 76 (4.1%) |
| Hospital readmissions | 160 (8.5%) |
| Re-operations | 107 (5.7%) |
| Total number of complication events | 355 |
| Number of people with a composite outcome | 234 (12.5%) |
| All joint/ surgical site infections (treated with oral / IV antibiotics, readmission, reoperation)* | 256 (13.7%) |
| Deep joint infections (treated with readmission or reoperation) | 63 (3.4%) |

*Participants with joint infection that required hospital readmission, or a reoperation were included in the composite outcome.

**Table 4. VTE, antibiotic and combined (composite) compliance.**

| Criteria for VTE compliance | Yes (N, %) |
|---|---|
| 1. Right drug (N = 1875) | 1518 (81%) |
| 2. Right dosage (N = 1860) | 1323 (70.6%) |
| 3. Right duration (Hip: ≥ 28 days, Knee: ≥ 10 days) (N = 1875) | 847 (45.2%) |
| 4. Right mechanical device / joint N = 15 | 1703 (90.8%) |
| Compliant with NHMRC (1, 2, 3, 4) (N = 1860) | 657 (35.3%) |
| **Criteria for antibiotic compliance** | **Yes (N, %)** |
| 1. Right drug (N = 1875) | 1418 (75.7%) |
| 2. Right dosage (including intra-op dose for op > 3hrs) (N = 1874) | 446 (23.8%) |
| 3. Right pre-op timing (any) (N = 1875) | 1780 (94.9%) |
| 4. Right duration (N = 1875) | 1025 (54.7%) |
| Compliant with TG: AB (Yes 1, 2, 3, 4) (N = 1874) | 247 (13.2%) |
| **Combined (composite) VTE and AB compliance (N = 1858)** | **84 (4.5%)** |

not statistically significant in either the unadjusted or adjusted analyses (See Table 5). There were no differences in any sensitivity analyses, with the effect estimate for composite non-compliance remaining non-significant when routine doppler ultrasound was removed.

**Association between VTE non-compliance and VTE outcomes.** There was a significantly higher incidence of VTE outcomes in the (VTE prophylaxis guideline) non-compliant group in both the unadjusted (p = 0.008) and adjusted analyses [5.0% vs 2.4%, AOR = 2.75, 95%CI 1.57–5.08, p < 0.001) (See Table 6). There were no meaningful differences in the effect estimate for VTE non-compliance (AOR = 2.62) or model results when routine doppler ultrasound was excluded from the model. There was no significant interaction between the main predictor (non-compliance) and other variables included in the final model and the other three analyses completed.

**Table 5. Association between composite non-compliance and composite outcome.**

| Unadjusted analyses | | | |
|---|---|---|---|
| Composite non-compliance (N = 1859) | No complications N (%) | Composite complications N (%) | p value (Chi-square) |
| Non-compliance | 1548 (87.3%) | 226 (12.7%) | 0.23 |
| Compliance | 77 (91.7%) | 7 (8.3%) | |
| Total | 1625 (87.3%) | 233 (12.8%) | |
| **Adjusted analyses–final regression model results** | | | |
| Variables in the final model | Adjusted OR (95% CI) | p value | |
| Composite non-compliance VTE and antibiotics clinical guidelines | 1.41 (0.68–3.45) | 0.40 | |
| Knee joint (TKR) | 2.15 (1.59–2.96) | <0.001 | *** |
| Comorbid Kidney disease | 2.01 (1.06–3.65) | 0.03 | * |
| Routine doppler | 1.86 (1.33–2.59) | < .001 | *** |
| Comorbid musculoskeletal condition | 1.44 (1.09–1.92) | 0.01 | * |
| 1st day mobilised day 0 or 1 | 0.73 (0.53–0.99) | 0.04 | * |
| Taking NSAIDs for arthritis | 0.66 (0.46–0.92) | 0.02 | * |
| Premorbid history of stroke | 1.51 (0.88–2.48) | 0.12 | |
| Comorbid GORD | 1.30 (0.95–1.76) | 0.10 | . |

Significance. codes

*** < 0.001

** <0.01

* <0.05.

**Table 6. Association between VTE non-compliance and VTE outcomes (final model).**

| Unadjusted analyses | | | |
|---|---|---|---|
| VTE non-compliance (N = 1860) | No VTE complications N (%) | VTE complications N (%) | p value (Chi-square) |
| Non-compliance | 1143 (95%) | 60 (5%) | 0.008 |
| Compliance | 641 (97.6%) | 16 (2.4%) | |
| Total | 1784 (95.9%) | 76 (4.09%) | |
| **Adjusted analyses–final regression model results** | | | |
| Variables in the final model | Adjusted OR (95% CI) | p value | |
| Non-compliance VTE clinical guidelines | 2.75 (1.57–5.08) | <0.001 | *** |
| Routine doppler | 4.00 (2.41–6.65) | <0.001 | *** |
| Knee joint (TKR) | 2.83 (1.59–5.28) | <0.001 | *** |
| Comorbid heart disease | 1.81 (1.05–3.07) | 0.03 | * |
| History previous VTE | 1.85 (0.85–3.67) | 0.09 | . |
| Comorbid musculoskeletal condition | 1.48 (0.91–2.42) | 0.11 | |
| Higher BMI | 1.03 (0.99–1.07) | 0.11 | |
| Increasing age | 1.03 (1.00–1.06) | 0.07 | . |
| Completed any post school education | 0.70 (0.43–1.14) | 0.15 | |
| 1st day mobilised day 0 or 1 | 0.66 (0.40–1.11) | 0.11 | |
| Higher ASA Score (3 or 4) | 0.65 (0.37–1.11) | 0.12 | |
| Comorbid depression or anxiety | 0.61 (0.28–1.18) | 0.17 | |

Significance. codes

*** < 0.001

** <0.01

* <0.05.

## Association between antibiotic non-compliance and all joint infection outcomes

There was a significantly higher incidence of joint infection outcomes in the (antibiotics guideline) non-compliant group in both the unadjusted (p = <0.001) and adjusted analyses (14.8% vs 6.1%, AOR = 1.98, 95%CI 1.17–3.62, p = 0.02) (See Table 7). VTE compliance was included in the SSI models as it was significant on bivariable analysis for both SSI outcomes and a biologically plausible confounder for infection. Non-compliance with VTE guidelines was also associated with all SSI outcomes (AOR = 1.52, 95%CI 1.11–5.42-, p < 0.01). There was no significant interaction between the main predictor (non-compliance) and other variables included in the final model.

## Association between antibiotic non-compliance and deep joint infection outcomes

The higher rate of deep infections in the non-compliant group (3.7%) than the compliant group (1.2%) was statistically significant in the unadjusted analysis but not in the adjusted analysis (3.7% vs 1.2%, AOR = 2.39, 95%CI 0.85–10.00, p = 0.15) (See Table 8).

Sensitivity analyses using a complete case analysis and using BIC model selection criteria did not demonstrate different results (See S1-S8 Tables in S1 File).

## Discussion

This is the first study to systematically examine the relationship between patient appropriate recommended care using Australian clinical guidelines for the prevention of VTE and

**Table 7. Association between antibiotic non-compliance and any Surgical Site Infection (SSI) outcome.**

| Unadjusted analyses | | | |
|---|---|---|---|
| Antibiotic non-compliance (N = 1872) | No SSI | Any SSI complications N (%) | p value (Chi-square) |
| Non-compliance | 1385(85.2%) | 241 (14.8%) | < .001 |
| Compliance | 232(93.9%) | 15 (6.1%) | |
| Total | 1616 (86.3%) | 256 (13.7%) | |
| **Adjusted analyses: Antibiotic compliance and all surgical site infections** | | | |
| Variables in the final model | Adjusted OR (95% CI) | p value | |
| Non-compliance antibiotic clinical guidelines | 1.98 (1.17–3.62) | 0.02 | * |
| Taking antiepileptic/antidepressant for arthritis | 2.54 (1.11–5.42) | 0.02 | * |
| Knee joint (TKR) | 2.40 (1.75–3.33) | <0.001 | *** |
| Comorbid neurological condition | 2.13 (1.05–4.06) | 0.03 | * |
| Non-compliance VTE clinical guidelines | 1.52 (1.13–2.05) | 0.01 | ** |
| Taking NSAIDs for arthritis | 1.37 (1.01–1.84) | 0.04 | * |
| Longer surgical duration | 1.01 (1.00–1.01) | <0.001 | *** |
| Higher BMI | 1.04 (1.02–1.06) | <0.001 | *** |
| Private hospital | 0.70 (0.51–0.95) | 0.02 | * |
| Bilateral joint replacement | 0.28 (0.11–0.60) | < .001 | ** |
| Premorbid history of stroke | 1.57 (0.92–2.57) | 0.09 | . |
| Comorbid sleep apnoea | 1.46 (0.91–2.29) | 0.11 | |
| Received blood transfusion | 1.32 (0.92–1.89) | 0.13 | |
| Comorbid lung disease | 0.72 (0.50–1.04) | 0.09 | . |

Significance. codes

*** < 0.001

** <0.01

* <0.05.

**Table 8. Association between antibiotic non-compliance and deep SSI outcomes.**

| Unadjusted analyses | | | |
|---|---|---|---|
| Antibiotic non-compliance (N = 1873) | No deep SSI | Deep SSI complications N (%) | p value (Chi-square) |
| Non-compliance | 1566 (96.3%) | 60 (3.7%) | 0.04 |
| Compliance | 244 (98.8%) | 3 (1.2%) | |
| Total | 1810 (96.6%) | 63 (3.4%) | |
| **Antibiotic compliance and deep surgical site infection outcomes** | | | |
| Variables in the final model | Adjusted OR (95% CI) | p value | |
| Non-compliance antibiotic clinical guidelines | 2.39 (0.85–10.00) | 0.15 | |
| Current smoker | 2.76 (1.32–5.33) | < .001 | * |
| Knee joint (TKR) | 2.70 (1.46–5.40) | < .001 | * |
| Higher BMI | 1.07 (1.03–1.10) | <0.001 | *** |
| Comorbid neurological condition | 2.48 (0.72–6.56) | 0.10 | . |
| Premorbid history of stroke | 2.22 (0.89–4.82) | 0.06 | . |

Significance. codes

*** < 0.001

** <0.01

* <0.05.

infection after elective THA and TKA and adverse patient outcomes. The study showed important and statistically significant associations between non- compliance with VTE prophylaxis guidelines and VTE complications, and non-compliance with antibiotic prophylaxis guidelines and increased risk of surgical site infection.

These results are consistent with one previous study that demonstrated increased risk of infection associated with non-compliance with the Therapeutic Guidelines Antibiotic (OR = 2.74, 95% CI 1.15–6.53), although this study also included patients undergoing THR for hip fracture [34]. Direct comparison with other studies is difficult due to variation in clinical guidelines used, the criteria for compliance, mixed surgical caseloads, limitations associated with administrative data and lack of follow-up in some studies [26, 35, 36]. These factors may explain the variation reported in previous studies. While some previous studies reported significant association between complications and adherence to evidence-based antibiotic [34, 37, 38] and VTE prophylaxis [39], other studies showed no significant difference in outcomes [23, 35, 40]. Unlike previous research, we did not observe a significant association between risk of complications and non-compliance with a composite process of care [26]. While the use of a composite measure aimed to reflect overall service quality, this may have masked the association between specific processes of care and outcomes [41, 42].

This study also demonstrated extremely high rates of non-compliance with both guidelines (95.6%), and VTE (63.5%) and antibiotic prophylaxis (86.7%) guidelines individually. The rate of VTE non-compliance was higher than other Australian studies (38–53% and 42% respectively) [18, 43]. The rate of antibiotic non-compliance reported in this study was much higher than recent Australian studies using the same antibiotic guidelines which reported 34% and 38.7% non-compliance [34, 44], but consistent with earlier Australian studies (86% and 86.7%) [43, 45]. In contrast, a Surgical Care Improvement Project study from the United States reported non-compliance rates as low as 4% for preventing infection and 2.5% for preventing VTE using administrative data [35]. International studies also reported wide variation in VTE and antibiotic guideline adherence, but again direct comparison remains difficult due to the use of different study methods, lack of adjustment for patient-appropriate variations and mixed patient populations [17, 46].

Our study also demonstrated high levels of unwarranted clinical variation in routine care to prevent infection and VTE after TKR and THR. The use of multiple prophylactic drugs, higher than recommended dose, and longer than recommended duration of prophylaxis, were surprising given the risks associated with inappropriate care [16, 47–50]. Prolonged VTE prophylaxis may increase the risk of bleeding; conversely when the duration or dose is insufficient, prophylaxis may be inadequate [50]. The high level of antibiotic overuse is concerning given the evidence suggesting that longer duration provides no additional protection against infection and has been associated with an increased risk of adverse events, antibiotic resistance, and higher cost of care [51, 52]. Nearly 60% of participants received more than one antibiotic when a single pre-operative dose of a single antibiotic is considered sufficient [12]. Inadequate prophylaxis due to underdosing is associated with higher risk of infections [53]. Given the risks associated with inappropriate care, further efforts to increase the implementation of evidence-based prophylaxis are urgently needed to improve the value and outcomes from THR and TKR [20, 21, 54].

This study has several limitations. Our sample was relatively consistent with national joint registry data for age, sex, body mass index (BMI) and ASA scores but included a higher proportion of THR and higher rate of surgery performed in public versus private hospitals [3]. Alternative approaches to determine compliance and use of different clinical guidelines may yield different results, however we considered our criteria to be more lenient than a strict interpretation of the guidelines. Further research should explore whether specific features of

prophylactic regimes have greater impact on reducing risk of complications. Replicating this study with current Australian and international guidelines for the prevention of VTE and surgical site infection is recommended [55].

A key strength of this study lies in the prospectively collected and audited clinical data and low rate of missing data that describes care provided in detail as opposed to large administrative data that may be prone to coding errors [56–58]. The current study allowed more comprehensive analyses of detailed processes of care from evidence-based clinical guidelines for the prevention of VTE and infection than previous studies [26, 59, 60]. We used robust criteria and computer-based algorithms for assessing compliant care that addressed patient-specific variations, as clinical decisions made in the patient's best interests may be deemed 'technically' non-compliant. The rigour of our approach and use of prospectively collected clinical data may help to explain the much higher rates of non-compliance we reported compared to most previous studies [9, 58].

The increased risk of VTE and infection complications associated with low levels of compliance with evidence-based prophylaxis suggest that further work to improve the implementation of clinical guidelines is needed. Reducing unnecessary clinical variation and preventing avoidable VTE and infection complications will improve the patient and service outcomes and should support cost-containment [20, 61]. Optimising patient outcomes from THR and TKR will improve the value and sustainability of these procedures.

## Supporting information

**S1 File. Additional results.**
(DOCX)

## Acknowledgments

Thanks to Dr Luan Dang, Shirley Cross and Carolyn Gray-Robens for verifying the accuracy of patient reported complications.

The authors would like to acknowledge the contribution of the following research assistants who supported data collection and data entry including Carolyn Gray-Robens, Shirley Cross, Marg Easterbrook, Michelle Jones, Nidhi Jain, Catherine Belousoff, Kelly Wheeler, and others.

In memory of Jo.

## Author Contributions

**Conceptualization:** Helen Mary Badge, Justine M. Naylor, John Fletcher, Iain Gosbell, Christine Lin, Ian A. Harris.

**Data curation:** Helen Mary Badge, Tim Churches, Justine M. Naylor, Wei Xuan, Elizabeth Armstrong, John Fletcher, Iain Gosbell.

**Formal analysis:** Helen Mary Badge, Tim Churches, Justine M. Naylor, Wei Xuan, Ian A. Harris.

**Funding acquisition:** Justine M. Naylor, Ian A. Harris.

**Investigation:** Helen Mary Badge, Justine M. Naylor, Leeanne Gray, Ian A. Harris.

**Methodology:** Helen Mary Badge, Tim Churches, Justine M. Naylor, Wei Xuan, Elizabeth Armstrong, John Fletcher, Iain Gosbell, Christine Lin, Ian A. Harris.

**Project administration:** Helen Mary Badge, Justine M. Naylor, Elizabeth Armstrong, Leeanne Gray, John Fletcher, Iain Gosbell, Christine Lin, Ian A. Harris.

**Resources:** Helen Mary Badge, Justine M. Naylor, Leeanne Gray, Ian A. Harris.

**Software:** Helen Mary Badge, Tim Churches.

**Supervision:** Tim Churches, Justine M. Naylor, Wei Xuan, Ian A. Harris.

**Validation:** Helen Mary Badge, Tim Churches, Justine M. Naylor, Ian A. Harris.

**Visualization:** Helen Mary Badge, Tim Churches.

**Writing – original draft:** Helen Mary Badge, Ian A. Harris.

**Writing – review & editing:** Helen Mary Badge, Tim Churches, Justine M. Naylor, Wei Xuan, Elizabeth Armstrong, Leeanne Gray, John Fletcher, Iain Gosbell, Christine Lin, Ian A. Harris.

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
