## [Decision Letter · Decision Letter 0]

11 Oct 2021

PONE-D-21-07165Non-compliance with clinical guidelines increases the risk of complications after total hip and knee joint replacement surgeryPLOS ONE

Dear Dr. Badge,

Thank you for submitting your manuscript to PLOS ONE. After careful consideration, we feel that it has merit but does not fully meet PLOS ONE’s publication criteria as it currently stands. Therefore, we invite you to submit a revised version of the manuscript that addresses the points raised during the review process. Please submit your revised manuscript by Nov 25 2021 11:59PM. If you will need more time than this to complete your revisions, please reply to this message or contact the journal office at plosone@plos.org. Please include the following items when submitting your revised manuscript:A rebuttal letter that responds to each point raised by the academic editor and reviewer(s). You should upload this letter as a separate file labeled 'Response to Reviewers'.A marked-up copy of your manuscript that highlights changes made to the original version. You should upload this as a separate file labeled 'Revised Manuscript with Track Changes'.An unmarked version of your revised paper without tracked changes. You should upload this as a separate file labeled 'Manuscript'.

We look forward to receiving your revised manuscript.

Kind regards,

Tim Schultz

Academic Editor

PLOS ONE

Additional Editor Comments (if provided):

Thank-you for your forebearance as this manuscript has moved between a number of Academic Editors.

The manuscript has now been reviewed by three reviewers. Reviewers 2 and 3 highlight minor points for feedback.

In your response to reviewers, please address items 1 and 3 from reviewer 2, and the comments from reviewer 3, in particular those addressing the broken field link.

Journal Requirements:

5. Thank you for stating the following in the Competing Interests section: "I have read the journal's policy and the authors of this manuscript have the following competing interests: 

JF is an unpaid member of the group that developed the NHMRC Guidelines for the prevention of VTE."

7. Your abstract cannot contain citations. Please only include citations in the body text of the manuscript, and ensure that they remain in ascending numerical order on first mention.

8. Please upload a copy of Figure 1, to which you refer in your text on pages 10 and 13. If the figure is no longer to be included as part of the submission please remove all reference to it within the text.

9. Please upload a copy of Supporting Information file 1,file 2, and Supplementary table 1-8 which you refer to in your text on page 37.

Reviewers' comments:

Reviewer's Responses to Questions

**Comments to the Author**

1. Is the manuscript technically sound, and do the data support the conclusions?

Reviewer #1: Yes

Reviewer #2: Yes

Reviewer #3: Yes

2. Has the statistical analysis been performed appropriately and rigorously? 

Reviewer #1: Yes

Reviewer #2: Yes

Reviewer #3: I Don't Know

3. Have the authors made all data underlying the findings in their manuscript fully available?

Reviewer #1: Yes

Reviewer #2: Yes

Reviewer #3: Yes

4. Is the manuscript presented in an intelligible fashion and written in standard English?

Reviewer #1: Yes

Reviewer #2: Yes

Reviewer #3: Yes

5. Review Comments to the Author

Reviewer #1: First, thank you so much for providing the opportunity to evaluate this work. This study gives important additional knowledge about complications after total hip and knee joint replacement surgery and need of compliance with evidence-based guidelines. The paper highlights a problem in a specific area which requires urgent attention. I congratulate the authors on the magnificent efforts and worthy research work. I think this work is acceptable for publication in your prestigious journal.

Reviewer #2: Review comments: PONE-D-21-07165

1. Does the manuscript provide a valid rationale for the proposed study, with clearly identified and justified research questions?

Reviewer comment:

This study is an important issue, but the authors have to clarify some statements. Are the compliance to the guidelines the only predictor factor to complication post-surgery?

Reference:

Schwartz FH, Lange J. Factors That Affect Outcome Following Total Joint Arthroplasty: a Review of the Recent Literature. Curr Rev Musculoskelet Med. 2017;10(3):346-355. doi:10.1007/s12178-017-9421-8.

Heo, S.M., Harris, I., Naylor, J. et al. Complications to 6 months following total hip or knee arthroplasty: observations from an Australian clinical outcomes registry. BMC Musculoskelet Disord 21, 602 (2020). https://doi.org/10.1186/s12891-020-03612-8.

2. Is the protocol technically sound and planned in a manner that will lead to a meaningful outcome and allow testing the stated hypotheses?

Reviewer comment:

Yes, it is

3. Is the methodology feasible and described in sufficient detail to allow the work to be replicable?

Reviewer comment:

The analysis methods isn’t clear. Logistic regression is used to describe data and to explain the relationship between one dependent binary variable and one or more nominal, ordinal, interval or ratio-level independent variables. This study: Logistic regression was undertaken to explore associations between non-compliance with:

(1) both VTE and antibiotic guidelines and the risk of composite surgical complications [all cause mortality, any VTE, readmission/reoperation for joint-related reasons (one-year) and non-joint-related reasons (35 days)], (2) VTE guidelines and VTE outcomes, (3) antibiotic guidelines and (3) all SSIs, and (4) deep SSIs (requiring readmission/reoperation).

This statement isn’t clear: Guideline non-compliance rates were high: 65% (VTE), 87% (antibiotics) and 95% for both guidelines. What ‘both guidelines’ mean? How 95% came from?

4. Have the authors described where all data underlying the findings will be made available when the study is complete?

The authors stated that all data are fully available without restriction.

5. Is the manuscript presented in an intelligible fashion and written in standard English?

Reviewer comment:

The authors must give attention to space between words in writing. There are some duplication sentences in the manuscript.

Reviewer #3: Thanks for submitting this interesting paper reviewing the association between VTE and SSI prophylaxis and patient outcomes. From the documentation provided the paper has undergone significant changes. The paper is well written and make a valuable contribution to the literature as there are relatively few papers that associate guideline adherence and patient outcomes.

I have a few comments regarding the paper, which are for clarity, as the aims, methods, results, and discussion are well linked and described. Note I am not a statistician and cannot comment on the rigor of the statistical analysis.

Throughout the manuscript there is a broken field link described as an error.

Abstract: please add the word "elective" to the description of the patient cohort in the methods.

The descriptions of the log regressions undertaken is confusing because it starts with "associations between non-compliance". A better description is provided in mid-page 11 that is simpler.

Intro: Add in the word primary to all the research questions. The term major adverse events is used in the first research question but complication is used in the rest of the manuscript. Please use one consistently.

Methods: telephone follow up with participants was performed but the pre-op data collection process was not described.

Discussion: I was unable to access the supplementary files and I am presuming that the data relating to the components of the guidelines that were reported as compliant or not are in these tables. I am checking as the 4th paragraph of the discussion is difficult to follow without this information.

6. PLOS authors have the option to publish the peer review history of their article (what does this mean?). If published, this will include your full peer review and any attached files.

Reviewer #1: No

Reviewer #2: **Yes: **Fauna Herawati

Reviewer #3: No

---

## [Author Response · Author response to Decision Letter 0]

29 Oct 2021

Reviewer #2 comments:

1. Does the manuscript provide a valid rationale for the proposed study, with clearly identified and justified research questions?

Reviewer feedback: 

This study is an important issue, but the authors have to clarify some statements. Are the compliance to the guidelines the only predictor factor to complication post-surgery?

Reference:

Schwartz FH, Lange J. Factors That Affect Outcome Following Total Joint Arthroplasty: a Review of the Recent Literature. Curr Rev Musculoskelet Med. 2017;10(3):346-355. doi:10.1007/s12178-017-9421-8.

Heo, S.M., Harris, I., Naylor, J. et al. Complications to 6 months following total hip or knee arthroplasty: observations from an Australian clinical outcomes registry. BMC Musculoskelet Disord 21, 602 (2020). https://doi.org/10.1186/s12891-020-03612-8.

Response: 

We acknowledge that there are many patient, surgeon and care factors associated with the risk of complications after total joint arthroplasty. The main aim of our study was to explore the association between non-compliance with the guidelines and complications, adjusted for likely confounders. The evidence on factors that may be associated with the outcomes informed data collection, and the variables assessed for inclusion in the models. We have listed many possible confounders (other causes or contributing factors) of complications including many of those listed in the references, and these data include many of those described in Table 2 including the participant characteristics ( socio-demographic information, past medical history, indications, and contraindications for VTE and antimicrobial prophylaxis), surgical (surgical duration, neuraxial anaesthesia, use of tranexamic acid etc) and acute processes of care (e.g. early mobilisation). In the results section, Tables 5-8 include the non-compliance and the variables that were retained in the final stepwise regression models for each analysis. To clarify this further we have included some additional text in the Data analyses section and references as suggested: 

Extract from abstract (additional words underlined): 

“Four adjusted logistic regression analyses were undertaken to explore associations between binary non-compliance and the risk of surgical complications.”

Word limit constraints prevent further elaboration in the abstract. 

Extract from manuscript (additional words underlined):

“We conducted four multiple logistic regression analyses to explore associations between non-compliance and risk of outcomes as follows: (1) non-compliance with both VTE and antibiotic guidelines and composite outcomes (2) VTE non-compliance and VTE outcomes, (3) antibiotic compliance and any SSI and (4) antibiotic compliance and deep SSI. Patient and care factors known to increase the risk of surgical complications (including VTE, SSI) were considered as potential confounders (30-32).”

30. Alamanda VK, Springer BD. The prevention of infection: 12 modifiable risk factors. Bone Joint J. 2019 Jan;101-B(1_Supple_A):3-9. doi: 10.1302/0301-620X.101B1.BJJ-2018-0233.R1. PMID: 30648488.

31. Zhang, Zh., Shen, B., Yang, J. et al. Risk factors for venous thromboembolism of total hip arthroplasty and total knee arthroplasty: a systematic review of evidences in ten years. BMC Musculoskelet Disord 16, 24 (2015). https://doi.org/10.1186/s12891-015-0470-0.

32. Schwartz FH, Lange J. Factors that affect outcome following total joint arthroplasty: a review of the recent literature. Curr Rev Musculoskelet Med. 2017;10(3):346-355. doi:10.1007/s12178-017-9421-8.”

3. Is the methodology feasible and described in sufficient detail to allow the work to be replicable?

Reviewer feedback: 

The analysis methods isn’t clear. Logistic regression is used to describe data and to explain the relationship between one dependent binary variable and one or more nominal, ordinal, interval or ratio-level independent variables. This study: Logistic regression was undertaken to explore associations between non-compliance with:

(1) both VTE and antibiotic guidelines and the risk of composite surgical complications [all cause mortality, any VTE, readmission/reoperation for joint-related reasons (one-year) and non-joint-related reasons (35 days)], (2) VTE guidelines and VTE outcomes, (3) antibiotic guidelines and (3) all SSIs, and (4) deep SSIs (requiring readmission/reoperation).

Response: 

We have amended the description of the logistic regression models undertaken. 

Extract from abstract (additional words underlined): 

“Four adjusted logistic regression analyses were undertaken to explore associations between binary non-compliance and risk of surgical complications: (1) composite (simultaneous) non-compliance with both VTE and antibiotic guidelines and composite complications [all-cause mortality, any VTE, readmission/reoperation for joint-related reasons (one-year) and non-joint-related reasons (35 days)], (2) VTE non-compliance and any VTE, (3) antibiotic compliance and any SSI, and (4) antibiotic compliance and deep SSI.”

Word limit constraints prevent further elaboration in the abstract. 

As stated above, we have listed many possible confounders (other causes or contributing factors) of complications including many of those listed in the references, and these data include many of those listed in Table 2, including the participant characteristics (Socio-demographic information, past medical history, indications , and contraindications for VTE and antimicrobial prophylaxis), surgical (surgical duration, neuraxial anaesthesia, use of tranexamic acid etc) and acute processes of care (e.g. early mobilisation). In the results section, Tables 5-8 include the non-compliance and the variables that were retained in the final stepwise regression models for each analysis. To clarify this further we have included some additional text in the Data analyses section and references as suggested: 

Extract from manuscript (additional words underlined): 

In the Methods (P10) we have already stated that compliance variables were binary:

“Compliance was assessed as a series of dichotomous variables for each element of the guideline, for patients who completed at least one follow-up “.

We have clarified this on Page 11 

“We conducted four multiple logistic regression analyses to explore associations between non-compliance and risk of outcomes as follows: (1) non-compliance with both (VTE and antibiotic) guidelines and composite outcomes (2) VTE non-compliance and VTE outcomes, (3) antibiotic non-compliance and any SSI and (4) antibiotic non-compliance and deep SSI. Patient and care factors known to increase the risk of surgical complications (including VTE, SSI) were considered as potential confounders (30-32).”

Reviewer feedback: 

This statement isn’t clear: Guideline non-compliance rates were high: 65% (VTE), 87% (antibiotics) and 95% for both guidelines. What ‘both guidelines’ mean? How 95% came from?

Response: 

The method section has been revised to clarify the methods that were undertaken and explain that compliance with both guidelines was required for an overall composite compliance as follows: 

Extract from abstract (additional words underlined): 

In the abstract we clarified that the first regression analysis explored:

(1) “composite (simultaneous) non-compliance with both (VTE and antibiotic) guidelines and composite complications”

and the following sentence has been revised:

“Guideline non-compliance rates were high: 65% (VTE), 87% (antibiotics) and 95% (composite compliance).”

Word limit constraints prevent further elaboration in the abstract. 

Extract from manuscript (additional words underlined): 

The first research question:

“ 1. Is there an association between simultaneous non-compliance with both (antibiotic and VTE) guidelines and the rate of surgical complications after elective primary total joint replacement surgery?”

In Table One we added 

Composite compliance Compliant with all VTE guideline elements 1 AND 2 AND 3 AND 4, and all antibiotic guideline elements 1 AND 2 AND 3 AND 4 

In the Methods (page 9) this is already described: 

“Composite (overall) compliance required prophylaxis that was compliant with all elements of both the VTE and the antibiotic guidelines.”

Results. In table 4 we added:

Combined (composite) VTE and AB compliance (N=1858) 84 (4.5%)

5. Is the manuscript presented in an intelligible fashion and written in standard English?

Reviewer feedback: 

The authors must give attention to space between words in writing. There are some duplication sentences in the manuscript.

Response: 

The spacing between words and duplication has been examined and corrected as necessary. 

Reviewer #3 comments: 

Reviewer feedback: 

1. Throughout the manuscript there is a broken field link described as an error.

Response: 

Apologies for this oversight; this wasn’t apparent in the word document I had for the previous submission. Captions have been changed to plain font and I have checked that all links are working prior to submitting the revised manuscript.

Reviewer feedback: 

2. Abstract: please add the word "elective" to the description of the patient cohort in the methods.

Response: 

The abstract methods have been revised as requested:

Extract from abstract (additional words underlined): 

“A prospective multi-centre cohort study was undertaken in consenting adults with osteoarthritis undergoing elective primary TKR/THR at one of 19 high-volume Australian public or private hospitals.”

Reviewer feedback: 

3. The descriptions of the log regressions undertaken is confusing because it starts with “associations between non-compliance”. A better description is provided in mid-page 11 that is simpler.

Response: 

As suggested, we’ve used the description of the logistic regression methods undertaken from mid page 11, with some amendments to ensure the abstract was within the word count limit as follows:

Extract from abstract (additional words underlined): 

“Four adjusted logistic regression analyses were undertaken to explore associations between non-compliance and risk of complications: (1) composite (simultaneous) non-compliance with both (VTE and antibiotic) guidelines and composite complications [all-cause mortality, VTE, readmission/reoperation for joint-related reasons (one-year) and non-joint-related reasons (35 days)], (2) VTE non-compliance and VTE, (3) antibiotic compliance and any SSI, and (4) antibiotic compliance and deep SSI.”

Word limit constraints prevent further elaboration in the abstract. 

Reviewer feedback: 

4. Intro: Add in the word primary to all the research questions. The term major adverse events is used in the first research question but complication is used in the rest of the manuscript. Please use one consistently.

Response: 

The word ‘primary’ has been added to the title and research question. The first research question revised, replacing major adverse events with surgical complications. 

Extract from manuscript (additional words underlined): 

Title: 

“Non-compliance with clinical guidelines increases the risk of complications after primary total hip and knee joint replacement surgery.”

Research questions:

“1. Is there an association between non-compliance with both (antibiotic and VTE) guidelines and the rate of surgical complications after elective primary total joint replacement surgery?

2. Is there an association between non-compliance with VTE prophylaxis guidelines and VTE after elective primary total joint replacement surgery?

3. Is there an association between non-compliance with antibiotic guidelines and post-operative infection rate (considering all SSI and deep SSI requiring readmission/reoperation) after elective primary total joint replacement surgery?” 

Reviewer feedback: 

5. Methods: telephone follow up with participants was performed but the pre-op data collection process was not described.

Response: 

The methods have been revised to clarify the methods used to collect preoperative information and clarify how all data were collected: 

Extract from manuscript (additional words underlined): 

“The site coordinators collected prospective pre-operative data from participants via interview, including socio-demographic information, past medical history, indications, and contraindications for VTE and antimicrobial prophylaxis. They collected acute care data from the medical records. Participants provided post-acute data via telephone follow-up at approximately 35, 90, and 365 days post-surgery. Participants and sites provided details regarding prophylaxis and surgical complications.”

Reviewer feedback: 

6. Discussion: I was unable to access the supplementary files and I am presuming that the data relating to the components of the guidelines that were reported as compliant or not are in these tables. I am checking as the 4th paragraph of the discussion is difficult to follow without this information.

Response: 

The data relating to the components of the guidelines that were reported as compliant or not are reported in Table 4. 

The supplementary files have been uploaded with the revised manuscript and are available. We were not able to include the data included in the Supplementary Tables in the main manuscript due to word count limitations. 

Additional requirements for resubmission 

Editorial feedback: 

Response: 

The file names have been revised accordingly and the files uploaded as part of this resubmission include: 

Current manuscript for further review

1. Manuscript revised (tracked changes)

2. Manuscript (clean)

Supplementary file

3. S1 file.doc

Editorial feedback: 

Response: 

The reference list has been checked to ensure it is complete and correct. None of the journal articles have been retracted.

Two references are no longer current but have been retained: 

The first has been rescinded but is directly relevant to the study: 

The NHMRC guideline (2009) were rescinded in 2016 and replaced with a clinical care standard (ACQSHC, 2020), rather than a clinical guideline. However, the NHMRC guidelines were current at the time the initial study commenced and were used for the initial evaluation of the association between guideline compliance and risk of VTE, so the reference has been retained. 

- Rescinded guideline: 

National Health and Medical Research Council. Clinical practice guideline for the prevention of venous thromboembolism (deep vein thrombosis and pulmonary embolism) in patients admitted to Australian hospitals. Melbourne: National Health and Medical Research Council; 2009.

- Current clinical care standard (however not relevant tostudy)

ACQSHC (2020). Venous Thromboembolism Prevention Clinical Care Standard. Sydney. ACQSHC. https://www.safetyandquality.gov.au/standards/clinical-care-standards/venous-thromboembolism-prevention-clinical-care-standard.

The second is a website that is no longer functioning but was source of information that informed site recruitment: 

The following website is no longer available but was the source of information to identify surgical volumes for site eligbility, so has been retained: 

Australian Government. My Hospitals 2012 [Available from: https://www.myhospitals.gov.au/.

The following changes have been made to references: 

- The Hamilton (2012) (previously reference 3) has been removed as it is not needed given other references included – this will help reduce word count 

- The AOA annual report (previously reference 4) has been updated from the 2019 to 2020 report to reflect the most recent information available and pre-COVID pandemic. 

- Mulder (2012) (previously reference 9) and Peel (2015) (previously reference 10) have been replaced with Springer (2017) and Shahi (2016)

- Removed AAOS (2012) reference as not needed

- Removed Runciman (2102) and Arecelus (2013 (previously reference 16-19)) – too old and replaced with a more recent systematic review - Farfan (2016),

- Remove Bull (2006) due to age and replaced with Leaper (Ab) 

- Removed second reference re Redcap (Harris, 2019) as only the one directly related to the software is relevant (Vanderbuilt University)

- (30-32) additional references added as per reviewer 2’s suggestions. We included the Schwartz (2017) reference recommended by reviewer 2 and included two systematic reviews (Zhang, 2019; Alamanda,2019) (as stated above in our response to reviewer 2’s feedback).

- Replaced Mulder (2012) and Bozic (2013) with a recent systematic review (Tan, 2019) for 

The high level of antibiotic overuse is concerning given the evidence suggesting that longer duration provides no additional protection against infection and has been associated with an increased risk of adverse events, antibiotic resistance, and higher cost of care.

- Removed Marshall and Elshaug (2013) (previously (66-68) for

Reducing unnecessary clinical variation and preventing avoidable VTE and infection complications will improve the patient and service outcomes and should support cost-containment.

- Removed Wang (2012) (more related to link with SSI and anticoagulation) as used same SCIP data and replaced with Gouvea and Bozic (2013), the latter which was used to inform the study design for this publication for

Direct comparison with other studies is difficult due to variation in clinical guidelines used, the criteria for compliance, mixed surgical caseloads, limitations associated with administrative data and lack of follow-up in some studies.

Editorial feedback: 

1. Please provide additional details regarding participant consent. In the ethics statement in the Methods and online submission information, please ensure that you have specified (1) whether consent was informed and (2) what type you obtained (for instance, written or verbal, and if verbal, how it was documented and witnessed). If your study included minors, state whether you obtained consent from parents or guardians. If the need for consent was waived by the ethics committee, please include this information.

Response: 

In the manuscript the consent process was clarified: 

“ Prior to data collection informed written consent was obtained from eligible participants and the signed consent form was witnessed by the site coordinator.”

This statement has been added to the “Ethics Statement” field of the submission form

Editorial feedback: 

Response: 

The DOI is now available and has been included in the revised manuscript: 

Extract from manuscript (additional words underlined): 

A deidentified version of the data set and the full R code for all analyses are available (https://doi.org/10.26190/mddw-by48).

Editorial feedback: 

7. Your abstract cannot contain citations. Please only include citations in the body text of the manuscript, and ensure that they remain in ascending numerical order on first mention.

Response: 

The abstract doesn’t include any citations, the numbers reflect the four separate logistic regression analyses that were undertaken. This has been reworded to clarify this: 

Extract from manuscript (additional words underlined): 

“Four adjusted logistic regression analyses were undertaken to explore associations between non-compliance and risk of complications: (1) composite (simultaneous) non-compliance with both (VTE and antibiotic) guidelines and composite complications [all-cause mortality, VTE, readmission/reoperation for joint-related reasons (one-year) and non-joint-related reasons (35 days)], (2) VTE non-compliance and VTE, (3) antibiotic compliance and any SSI, and (4) antibiotic compliance and deep SSI.” 

Editorial feedback: 

8. Please upload a copy of Figure 1, to which you refer in your text on pages 10 and 13. If the figure is no longer to be included as part of the submission please remove all reference to it within the text.

Response: 

Figure 1 was included in the body of the manuscript and has now been uploaded as a separate file with the revised manuscript. 

Editorial feedback: 

9. Please upload a copy of Supporting Information file 1,file 2, and Supplementary table 1-8 which you refer to in your text on page 37.

Response: 

Supplementary file 3 is now Supplementary file 1 and includes table 1-8. This has been uploaded with the revised manuscript (See S1_file).

 The references to Supplementary files 1 (page 10) and 2 (page 12) have been removed as these are now available online. 

The last sentence in the methods section of the original submission has been deleted: The full R code for all analyses are included in a supplementary file (Supplementary File 2). 

Extract from manuscript (additional words underlined): 

New information has been added: 

“A deidentified version of the data set and the full R code for all analyses are available (https://doi.org/10.26190/mddw-by48).”

---

## [Editor Report · Decision Letter 1]

4 Nov 2021

Non-compliance with clinical guidelines increases the risk of complications after total hip and knee joint replacement surgery

PONE-D-21-07165R1

Dear Dr. Badge,

We’re pleased to inform you that your manuscript has been judged scientifically suitable for publication and will be formally accepted for publication once it meets all outstanding technical requirements.

Kind regards,

Tim Schultz

Academic Editor

PLOS ONE

Additional Editor Comments (optional):

Thank-you for your thorough revision of this manuscript in light of reviewer and editorial feedback.

The suggested revisions address the feedback appropriately.

Two minor suggestions for revision for the final manuscript:

1. line 47 (abstract) states "(2) VTE non-compliance and VTE", this could be clarified to "VTE non-compliance and VTE outcomes" as occurs later in the text, or "VTE non-compliance and any VTE" as per other outcomes.

2. the broken field link has recurred in this version; suggest just overwrite this with text rather cross-references in MS-Word

Lastly, the DOI link for the R analysis did not work https://doi.org/10.26190/mddw-by4
---

## [Editor Report · Acceptance letter]

9 Nov 2021

PONE-D-21-07165R1 

Non-compliance with clinical guidelines increases the risk of complications after primary total hip and knee joint replacement surgery 

Dear Dr. Badge:

I'm pleased to inform you that your manuscript has been deemed suitable for publication in PLOS ONE. Congratulations! Your manuscript is now with our production department. 

Kind regards, 

on behalf of

Dr. Tim Schultz 

Academic Editor

PLOS ONE